# Clinical Characteristics and Genetic Variants of a Large Cohort of Patients with Retinitis Pigmentosa Using Multimodal Imaging and Next Generation Sequencing

**DOI:** 10.3390/ijms241310895

**Published:** 2023-06-30

**Authors:** Richard Sather, Jacie Ihinger, Michael Simmons, Tahsin Khundkar, Glenn P. Lobo, Sandra R. Montezuma

**Affiliations:** Department of Ophthalmology and Visual Neurosciences, University of Minnesota, Minneapolis, MN 55455, USA; sathe130@umn.edu (R.S.III); jacqueline.ihinger@fairview.org (J.I.); simmo720@umn.edu (M.S.); tkhundkar9@gmail.com (T.K.); lobo0023@umn.edu (G.P.L.)

**Keywords:** retinitis pigmentosa, genetic testing, next generation sequencing, inheritance patterns, ocular coherence tomography, fundus autofluorescence

## Abstract

This retrospective study identifies patients with RP at the Inherited Retinal Disease Clinic at the University of Minnesota (UMN)/M Health System who had genetic testing via next generation sequencing. A database was curated to record history and examination, genetic findings, and ocular imaging. Causative pathogenic and likely pathogenic variants were recorded. Disease status was further characterized by ocular coherence tomography (OCT) and fundus autofluorescence (AF). Our study cohort included a total of 199 patients evaluated between 1 May 2015–5 August 2022. The cohort included 151 patients with non-syndromic RP and 48 with syndromic RP. Presenting symptoms included nyctalopia (85.4%) photosensitivity/hemeralopia (60.5%), and decreased color vision (55.8%). On average, 38.9% had visual acuity of worse than 20/80. Ellipsoid zone band width on OCT scan of less than 1500 μm was noted in 73.6%. Ninety-nine percent had fundus autofluorescence (AF) findings of a hypo- or hyper-fluorescent ring within the macula and/or peripheral hypo-AF. Of the 127 subjects who underwent genetic testing, a diagnostic pathogenic and/or likely pathogenic variant was identified in 67 (52.8%) patients—33.3% of syndromic RP and 66.6% of non-syndromic RP patients had a diagnostic gene variant identified. It was found that 23.6% of the cohort had negative genetic testing results or only variants of uncertain significance identified, which were deemed as non-diagnostic. We concluded that patients with RP often present with advanced disease. In our population, next generation sequencing panels identified a genotype consistent with the exam in just over half the patients. Additional work will be needed to identify the underlying genetic etiology for the remainder.

## 1. Introduction

Retinitis pigmentosa (RP) is the most common inherited retinal dystrophy (IRD) and is associated with progressive night vision loss, visual field constriction, reduced electroretinographic responses, and reduction in visual acuity [1]. The metabolic abnormalities associated with RP affect the rod and cone photoreceptors of the retina. On clinical exam, the characteristic phenotype consists of mid-peripheral retinal pigmentary changes (bone spicules), arteriolar attenuation, and waxy disc pallor, but it is a heterogenous group of disorders with at least 80 causative genes [2]. Although most patients with RP present with isolated eye findings, 20–30% of patients present with syndromic RP with multiorgan involvement [3].

RP can be transmitted by autosomal dominant, autosomal recessive, or X-linked inherence patterns. In the literature, the autosomal dominant inheritance pattern accounts for 20–25% of RP; an autosomal recessive pattern is observed in 15–20%, an X-linked pattern in 10–15%, and sporadic/simplex traits are observed in 30% [3]. The distribution of gene prevalence varies based on the population studied. For example, a Japanese population study of 68 patients found that one-third of patients with non-syndromic autosomal recessive RP carried pathogenic gene variants in the *EYS* gene [4], while a similar study from a western European ancestry cohort approximated that the prevalence of *EYS* variants accounted for only 5% of autosomal recessive RP in a cohort of 245 patients [5].

An estimated 20–30% of those diagnosed with RP are classified as syndromic patients [3]. The most common syndromic form of RP is Usher syndrome, which accounts for about 14% of all RP patients [6]. Usher syndrome belongs to a group of ciliopathies that causes defects in the ciliary protein trafficking. This condition usually affects multiple organ systems because numerous cells in the body, including the photoreceptors, have cilia. Usher syndrome is inherited in an autosomal recessive inheritance pattern and is characterized by congenital deafness and adolescent onset rod–cone dystrophy. There are three types of Usher syndrome. Each type displays a severity of deafness and variable vestibular response. Type 1 is the most severe, while type 2 and 3 are milder forms with a later onset of retinal degeneration. The next most frequent syndromic condition is Bardet Biedl syndrome, at a prevalence of 1/150,000 [7]. This autosomal recessive ciliopathy is characterized by rod–cone dystrophy, obesity, polydactyly, varying degrees of cognitive impairment, and genitourinary and renal abnormalities. Other syndromic RP diseases are subdivided into those that manifest with renal abnormalities, dysmorphic syndromes, metabolic diseases, or neurological diseases. Studying the mechanisms involved in facilitating and maintaining proper protein transport in photoreceptor cells will help to identify the underlying pathology of retinal cell degeneration for many of these conditions [8].

Identifying genotype–phenotype correlations in non-syndromic RP can be challenging because variants in different parts of the same gene can result in different phenotypes. For instance, variants in the *ABCA4* gene, which encodes an ATP-binding cassette transported expressed in the disc of photoreceptor outer segments, can have several phenotypes. These include Stargardt disease, fundus flavimaculatus, RP, and cone–rod dystrophy. The different phenotypes reflect the different tissues in which the ABCA4 gene is expressed, including the photoreceptors and the retinal pigment epithelium [9,10].

Another example of a gene causing different phenotypes includes the peripherin-2 (*PRPH2*) gene, which encodes a photoreceptor-specific tetraspanin protein called peripherin-2. This protein is involved in membrane fusion and is required for the formation and maintenance of the outer segments of rods and cones. Its phenotypes include retinitis pigmentosa and macular degeneration [11].

An important imaging biomarker for monitoring progression of structural damage in RP is the width of the preserved ellipsoid zone (EZ) within the macula [12]. Studies using optical coherence tomography (OCT) to measure the EZ have extended the use of EZ width to distinguishing disease progression in differing inheritance patterns of RP. It has been shown that the mean rate of decline in EZ width of 7% represents a mean rate of change of 13% for the equivalent area of the EZ [13]. This rate of change is similar to findings reported for Goldmann visual fields and full-field electroretinograms [14,15]. Along with fundus autofluorescence (FAF) and Goldmann visual fields, EZ band width measurements are now being incorporated into inclusion/exclusion criteria to limit participants with advanced disease severity in gene therapy clinical trials.

FAF has been proposed as an indirect biomarker marker of RPE function and can also be used to assess RP progression. FAF highlights the distribution of fluorophores in the RPE, such as lipofuscin and lipofuscin accumulation. Specific patterns of increased fluorescence are suggestive of oxidative stress and increased metabolic activity. Hyperautofluorescence in the macula indicates RPE stress, whereas hypoautofluorescence can indicate RPE loss. Genotype correlations with FAF phenotypes have been investigated using ultra widefield fundus autofluorescence (UW FAF) in a cohort of patients. Meaningful FAF patterns included a ring of hyperautofluorescence, double ring hyperautofluorescence, and peripheral hypoautofluorescence [16].

There are several therapeutic trials underway for the treatment of IRDs [17]. Patients with biallelic variants in *RPE65* now have a commercially available gene-replacement treatment [18]. With more targeted gene therapy treatments likely to be available in the future, it is critical to describe this class of disorders on a genetic level [19]. The IRD service at the University of Minnesota is a referral center for the state and neighboring regions. This study reports the pathogenic gene variants found utilizing next generation sequencing (NGS) in individuals in our population with syndromic and non-syndromic RP. We report patient clinical characteristics along with the gene variants found in this cohort.

## 2. Results

### 2.1. Demographic Information

A total of 199 patients diagnosed with RP were found within the evaluated time frame per our IRB. The patient distribution consisted of 151 patients with non-syndromic RP and 48 with syndromic RP. The syndromic RP patients included Usher syndrome (39), Bardet Biedl syndrome (4), Cohen syndrome (2), nephronophthisis (1), cardiofaciocutaneous syndrome (1), and abetaproteinemia (1). There were 98 males (49.2%) and 100 females (50.2%). One patient with an XY chromosomal arrangement did not identify with a binary gender. The age range of the cohort include the following: before the age of 10 (4), between the ages of 10–19 (21), between the ages of 20–40 (47), and after the age of 40 (127). The following observed symptoms were recorded from the baseline evaluation. In terms of age range for the initial genetic testing: before the age of 10 (11), between the ages of 10–19 (24), between the ages of 20–40 (62), and after the age of 40 (102).

### 2.2. Symptoms

A total of 61 of the 144 (42.4%) subjects first noted eye symptoms before the age of 10. A total of 68 of the 121 (56.2%) subjects had a known family history of retinal dystrophy (Table 1). A total of 134 of the 157 (85.4%) reported nyctalopia, 52/86 (60.5%) reported photosensitivity and or hemeralopia, 170/184 (92.4%) reported visual field loss, and 53/95 (55.8%) reported color vision impairment measured by the Ishihara test. The varying denominators reflect the number of subjects for whom this information was available through retrospective chart review.

### 2.3. Visual Acuity Results

A total of 74 of the 198 (37.4%) subjects had visual acuity worse than 20/80 in the right eye and 80/198 (40.4%) for the left eye. There was not a statistically significant difference found between the two eyes.

### 2.4. Ellipsoid Zone (EZ) Measurements

EZ measurements were taken at the baseline visit. Advanced photoreceptor loss, representing less than 1500 µm, was noted in 134/182 (73.6%) of subjects. EZ band width measurement (see Table 1) on a foveal OCT scan is illustrated (Figure 1A).

### 2.5. Fundus Autofluorescent (FAF) Patterns

Almost all patients—191/193 (99.0%)—presented with FAF findings of either a ring of macular hypo/hyper AF or peripheral hypo-AF in at least one eye. Figure 1B illustrates FAF findings in our patient cohort and our criteria to distinguish between imaging findings of macula hypo/hyper AF ring or peripheral hypo-AF. Both EZ band width ranges and FAF findings between left and right eye are reported in the demographic section in Table 1.

### 2.6. Genetic Testing Reports

A summary of the commercial genetic panels that our patient cohort used, and their corresponding diagnostic yield rate is presented in Table 2.

A total of 127/199 (63.8%) patients had genetic testing completed at the time of this study. At least one pathogenic variant was identified in 97/127 (78.0%) patients. Of these, 67/127 (52.8%) had a pathogenic variant that was diagnostic (Figure 2(1a)). This percentage reflects the diagnostic yield for our RP patient cohort.

There were 60/127 (47.2%) patients who underwent genetic testing and did not have a diagnostic pathogenic variant identified (Figure 2). The majority (22/127 (17.3%)) had only VUS identified, and 8/127 (6.3%) had negative results (Figure 2 (column 2)). For the remaining 30 patients (Figure 2 (columns 1b and 1c)), a pathogenic variant was identified, but was not diagnostic for two reasons. First, for the 15/127 patients (Figure 2 (column 1b)), only one pathogenic/likely pathogenic variant was found, while the second variant was classified as VUS. As previously mentioned in the methods section, both alleles must be classified as pathogenic or likely pathogenic and confirmed or presumed to be in the trans configuration to be considered for diagnostic criteria of autosomal recessive RP. Second, the other 15/127 patients (Figure 2 (column 1c)) had only a single pathogenic variant identified in an autosomal recessive gene, indicating carrier status. Figure 2 provides a full genetic summary of the patients that underwent genetic testing.

In our patient cohort, 97 pathogenic/likely pathogenic variants were identified in 32 different genes associated with syndromic and non-syndromic RP phenotypes. The genes identified for each diagnostic syndromic and non-syndromic RP are also listed within Figure 2 (column 1a). The top identifiable genes included *RPGR* (16%), *USH2A* (12%), *MYO7A* (12%), *RP1* (6%), *EYS* (4%), *PRPH2* (4%), *BBS1* (4%), *PDE6B* (3%), and *VPS13B* (3%) (Figure 3). The variants identified in each individual case, zygosity, and ACMG classification are available in the Appendix A.

### 2.7. The Inheritance Pattern Distribution

Our cohort included 12/67 (17.9%) autosomal dominant, 43/67 (64.2%) autosomal recessive, and 12/67 (17.9%) X-linked RP (Figure 2). Within these inheritance patterns, 11/12 (91.7%) patients with X-linked RP had a diagnostic *RPGR* variant. Nine of those patients were male, and one patient was a heterozygous female with both of her sons having a more severe disease presentation. For the patients with autosomal dominant RP, four had the *RP1* variant and a family of three had the *PRPH2* variant.

### 2.8. Supplemental Data

The Appendix A contains details on which variants were identified in each individual patient and is written in accordance with the current Human Genome Variation Society (HGVS) nomenclature, the heterozygosity of variants, and the American College of Medical Genetics (ACMG) variant classifications, as assigned by the performing genetic laboratory. The data in this database were extracted from genetic test reports from various Clinical Laboratory Improvement Amendments (CLIA) and accredited by the College of American Pathologists (CAP) certified laboratories. We did not perform specific splice predictions and/or in silico predictions.

## 3. Methods

Our cohort of individuals with syndromic and non-syndromic RP was studied retrospectively. The patients included were all evaluated at the IRD Clinic at the UMN/M Health System. All patients seen between 1 May 2015 (the date our institution implemented its current electronic medical record system) and 5 Aug 2022, were included according to our IRB STUDY00012478. The collected data did not exclude patients based on age, race, or gender. Patients within our hospital system may opt out of inclusion in retrospective chart reviews at the time of initial consent for service. All patients who opted out were excluded from this analysis.

### 3.1. Database

Clinical information was collected using our institution’s electronic healthcare record system. The REDCap software platform was used to curate a database. An original survey was constructed to facilitate retrospective collection of demographic, history, and exam findings for each patient through EPIC. Each patient received a randomized numerical assignment, accompanied by their medical identification number. Any question addressed in the survey that was not directly found in the patient chart was labeled as ‘unknown’.

The data entry included questions regarding present ocular history, family history of retinal degeneration, baseline ocular examination, genetic report, and diagnostic imaging. In addition, the age at which the patient first noted eye symptoms was recorded. These ranges include <10, 10–19, 20–40, and >40 years of age. The data entry for baseline ocular examination included the presence of nyctalopia, hemeralopia/photosensitivity, visual acuity, and visual field loss. Visual acuity was classified as being either 20/40 or better, worse than 20/40 but better than or equal to 20/80, or worse than 20/80. The presence of visual field loss was analyzed based on results of the Goldmann visual field testing.

Multimodal imaging for patients in our cohort included FAF (Optos^®^) and OCT imaging (Heidelberg-Spectralis^®^). These are the instruments that we have available in the clinic. These devices are commonplace in many retina clinics. The FAF demonstrated the presence of hypo- vs. hyper autofluorescence in the macula and peripheral retina. OCT imaging was used to analyze macular ellipsoid zone (EZ) band width. The marking of the EZ endpoint locations was measured manually. Two graders, including one retina specialist, evaluated EZ band width for all patients. Subjects with an EZ band width of less than 1500 um were considered to have advanced photoreceptor loss.

### 3.2. Genetic Testing

Genetic testing was offered to all patients during the IRD evaluation. Whether genetic testing was performed was noted along with the genetic variant(s) for each patient. The number of genes analyzed varied from single-gene targeted testing to panels with >300 genes. Single gene targeted testing was often employed when there was a known familial variant previously identified. The majority of patients had genetic testing performed via a next-generation sequencing (NGS) inherited retinal disease panel. The exact number of genes on these panels varied based on when the patient had genetic testing performed. Over the course of the study, the commercial providers of the NGS panels incorporated additional genes so patients evaluated at the end of the study period had more genes tested than those at the beginning.

In the past, common reasons for patients choosing not to undergo genetic testing included cost concerns, lack of interest, concern for too much information being requested, or personal preference. Four of the most common genetic testing laboratories that were used included Invitae Laboratory, Blueprint Genetics, PreventionGenetics, and the University of Minnesota Molecular Diagnostic Laboratory. These genetic testing laboratories are accredited College of American Pathologists (CAP) and Clinical Laboratory Improvement Amendments (CLIA) certified and utilize the currently available American College of Medical Genetics and Genomics (ACMG) variant classification guidelines to classify each variant identified. The percentage of patients who utilized each gene panel was considered, and an analysis of the pathogenic/likely pathogenic gene diagnostic yield rate was calculated for each of the listed gene panels. The presence of variants of uncertain significance (VUS) was also recorded for each patient.

The use of NGS has been shown to be an effective method for detecting pathogenic gene variants [20]. There are few possible outcomes of genetic testing: pathogenic/likely pathogenic variants for genes known to cause the phenotype in question, pathogenic/likely pathogenic variants for genes associated with phenotypes the patient does not have, variants of uncertain significance (VUS), or no variants. NGS panel reports consist of a list of genetic variants identified in the patient sample that could be associated with an IRD.

Our approach for determining whether a patient’s genetic testing results were diagnostic of RP aligns with the ACMG standards and guidelines for the interpretation of sequence variants [21]. The majority of patients who obtained genetic testing met with both a genetic counselor (JI) and a vitreoretinal specialist (SM). A clinical history and family history was collected for each patient.

Patients were identified as having diagnostic genetic results if (1) the patient had a sufficient number of pathogenic/likely pathogenic genetic variant(s), (2) the genetic variants were consistent with the patient phenotype, and (3) the genetic variants were consistent with the known inheritance pattern. For autosomal dominant inheritance patterns, only one likely pathogenic or pathogenic variant was required. For autosomal recessive conditions, two pathogenic or likely pathogenic variants were needed. When possible, family studies were conducted to confirm the two variants were in the trans configuration (i.e., one variant on each allele). In other instances, the variants were confirmed to be in trans based on sequencing results, homozygosity for the variant, or they were presumed to be in trans based on the patient’s clinical phenotype. If a single pathogenic variant was identified but that gene was associated with autosomal recessive inheritance, then the patient was classified as a carrier. There were situations where testing identified a single pathogenic/likely pathogenic variant in addition to a single VUS in the same autosomal recessive gene. These cases were deemed clinically suspicious but not as diagnostic for the purposes of this study. A patient was considered to have negative genetic test results if no genetic variants were identified or if the only variants identified were VUS.

## 4. Discussion

Of the 199 patients diagnosed with a RP phenotype, 127 patients underwent genetic testing, and 32 different genes were identified as the causative gene for the patients’ RP (Figure 3). Using NGS, we achieved a diagnostic yield of 52.8%, in which a pathogenic or likely pathogenic variant(s) was identified that was determined to be causative in the patient’s diagnosis. This diagnostic yield is higher than some reports in literature (~30–40%) [22,23], nearly identical with one (53.2%) [24], and below others (~60–70%) [25,26] that utilized NGS.

Molecular genetic testing is essential in the phenotypic diagnosis for patients with syndromic and non-syndromic RP. Like many clinical sites, the UMN IRD clinic utilizes a variety of commercial NGS gene laboratories for their patients (Table 2). The exact decision of which NGS panel or laboratory to use was based on several factors, including but not limited to the number of genes analyzed on a panel, family member testing (i.e., if a known familial variant was being testing, use of the same genetic testing laboratory was preferred), ease of sample collection, insurance coverage for testing, or a patient electing to participate in a sponsored testing program. The most frequently used gene panel for our patient cohort came from Invitae Laboratory. A total of 71 of 127 (55.9%) patients underwent genetic testing through this gene panel with a diagnostic yield rate of 43.7%. Of the 67 patients who had a diagnostic test result, 31 of those patients had testing at Invitae Laboratory. Yet, this is where most of our panels were sent. Other gene panels utilized were from the UMN Molecular Diagnostic Laboratory, PreventionGenetics, and Blueprint Genetics. Any additional genetic panel that was not one of these ones listed came from other medical institutions and their respective gene laboratory panel. The wide range in the diagnostic yield rate from the various genetic testing laboratories (Table 2) may be explained by the difference in the number of genes analyzed at the time or if several tests were sent to one genetic testing laboratory due to a known familial variant being previously identified at that laboratory. However, the diagnostic yield rate for any of these mentioned gene panels was 50% or greater. The wide range in the number of genes tested for different patients somewhat limits the conclusions for our study.

The most common causative genes in our patient cohort were *RPGR* (16%), *USH2A* (12%), *MYO7A* (12%), *RP1* (6%), *EYS* (4%), *PRPH2* (4%), *BBS1* (4%), *PDE6B* (3%), and *VPS13B* (3%) (Figure 3). Together, the diagnostic yield rate of 52.8% supports the use of NGS as an effective tool for RP patient diagnosis. Our cohort consisted of 151 non-syndromic and 48 (24.1%) syndromic RP patients. Twenty-three (47.9%) of the 48 syndromic RP patients had a diagnostic pathogenic variant. The distribution of the syndromic RP conditions is in the range of a report from a European cohort, which found a causative variant(s) in 59% of their syndromic cases [27] and higher than that of a Danish cohort, which found causative variant(s) in 28% of syndromic RP patients [28]. In addition, our percentage of total diagnostic Usher syndrome patients (65.2%) was larger than the Danish cohort, which consisted of only 43% of the total syndromic cases. The proportion of syndromic RP patients was additionally higher than reported in a Spanish cohort, which reported 18% syndromic RP patients [29].

Approximately two-thirds of our RP patients with a diagnostic pathogenic/likely pathogenic gene variant were considered non-syndromic. These consisted of 12 autosomal dominant (17.9%), 43 autosomal recessive (64.2%), and 12 X-linked RP (17.9%) (Figure 2). In agreement with the first published series from the United States (a 1978 study of 173 RP patients), the percent of our cohort with X-linked RP is relatively high. This higher percentage of X-linked RP was corroborated in the Denmark study [28] but was higher than other RP studies [30,31,32,33]. It should be noted that some our X-linked RP patients consisted of members of the same family.

Patients who had non-diagnostic results were represented in Figure 2(column 1b + 1c and column 2). For the 15 patients with one pathogenic/likely pathogenic variant and one VUS identified who were categorized in column 1b, the results for follow-up family studies were not available to confirm the configuration (cis or trans) of the two variants. The other 15/97 patients (Figure 2(column 1c)) had a single pathogenic variant identified in an autosomal recessive gene. In the future, it may be that the diagnostic yield rate of NGS panels will increase significantly as new variants and/or genes are discovered. However, it is possible that a portion of the non-diagnostic results in our cohort reflect non-genome-encoded epigenetic processes, such as post-transcriptional modification. For example, Donato et al. recently demonstrated significant post-transcriptional RNA editing activity in a model of cultured human-derived retinal pigment epithelial cells subjected to oxidative stress with N-retinylidene-N-retinyl ethanolamine (A2E) [34]. In the future, perhaps assays of RNA sequences or other assays of epigenetic influences may supplement NGS and improve the overall diagnostic yield of testing [35].

Whole exome sequencing (WES) or whole genome sequencing (WGS) may also prove valuable in investigating patients with negative NGS panel results [36]. Evaluating the diagnostic efficacy of this approach for patients with RP may be a direction for future work [37].

In our cohort, 73.6% of patients had advanced photoreceptor loss indicated by an EZ band width on the fovea scan of <1500 μm in at least one eye. Likewise, 99% of patients presented with FAF findings of either ring of macular hypo/hyper-AF or peripheral hypo-AF in at least one eye. Such an advanced-stage disease presentation is concerning as it may exclude patients from ongoing clinical trials. Although 127 patients first reported eye symptoms before the age of 40, 102 patients did not have their initial genetic test until after the age of 40. Age could then be a potential factor in advanced photoreceptor loss, as demonstrated by the OCT EZ band width interpretation. When patients present with a phenotype that is strongly suggestive of RP, obtaining genetic testing at the initial visit may enable them to identify clinical trials and potential future treatments at the earliest possible disease stage.

## 5. Limitations

One limitation of this study is that genetic testing was performed using NGS without utilizing WGS. Additionally, patients in our cohort who had testing early in the study window typically had fewer genes tested than those who were tested more recently. As more genes are discovered and additional genetic testing becomes available, patients should be educated about the potential for future updates of gene panels and additional testing. This same concept also applies to family members of affected individuals who were unavailable or chose not to be tested to provide further interpretation of a VUS via segregation analysis. The need for increased information of current therapeutic treatments should be provided to all patients who are diagnosed with an inherited retinal disease. In addition, these laboratories occasionally provide updated variant classifications on the same genetic testing as they gather more knowledge and evidence, particularly for VUS. This dynamic process makes providing definite conclusions regarding the gene prevalence in a population more challenging.

## 6. Conclusions

In our cohort, a diagnostic pathogenic/likely pathogenic variant was identified in 52.8% of patients who underwent genetic testing. Of these, 33.3% corresponded to syndromic RP and 66.6% to non-syndromic RP patients. The numerous patients with non-diagnostic results (i.e., identified as an asymptomatic carrier of an autosomal recessive IRD or VUS) suggests the need for a more encompassing genetic analysis and to promote testing for other affected family members to determine if the gene variant is pathogenic. A comprehensive approach involving genetic counselling, clinical evaluation, and appropriate imaging is necessary to properly characterize patients with this heterogenous disease. RP patients evaluated at UMN often present with advanced features of their disease. With potential future treatments for IRD, including gene therapy, stem cell, neuroprotection, retinal implants and optogenetics, prompt diagnosis may help identify subjects that could qualify for currently approved gene replacement therapies and ongoing or future clinical trials.

## Figures and Tables

**Figure 1 ijms-24-10895-f001:**
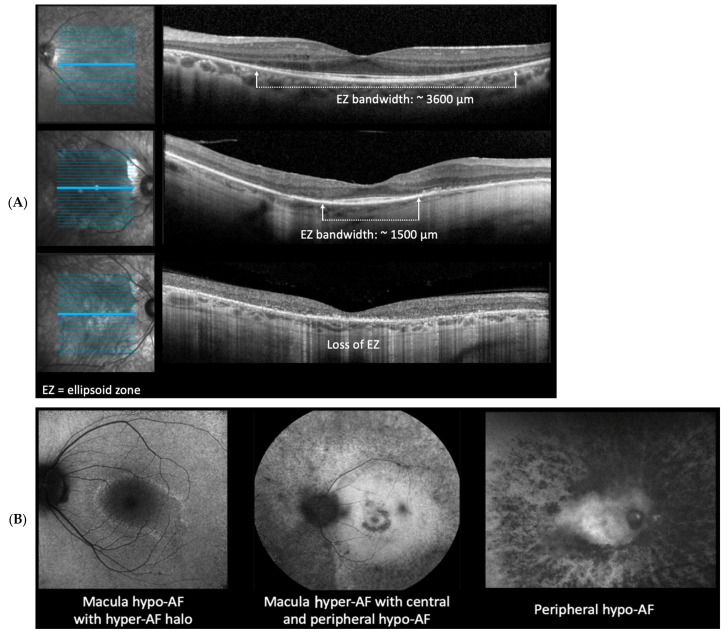
Criteria for diagnostic imaging modalities utilized for the patient cohort. (**A**) example parameter measurement of patient OCT EZ (ellipsoid zone) bandwidth interpretation. (**B**) Example criteria for FAF (fundus autofluorescence) imaging. Macula hypo/hyper-AF and peripheral hypo-AF.

**Figure 2 ijms-24-10895-f002:**
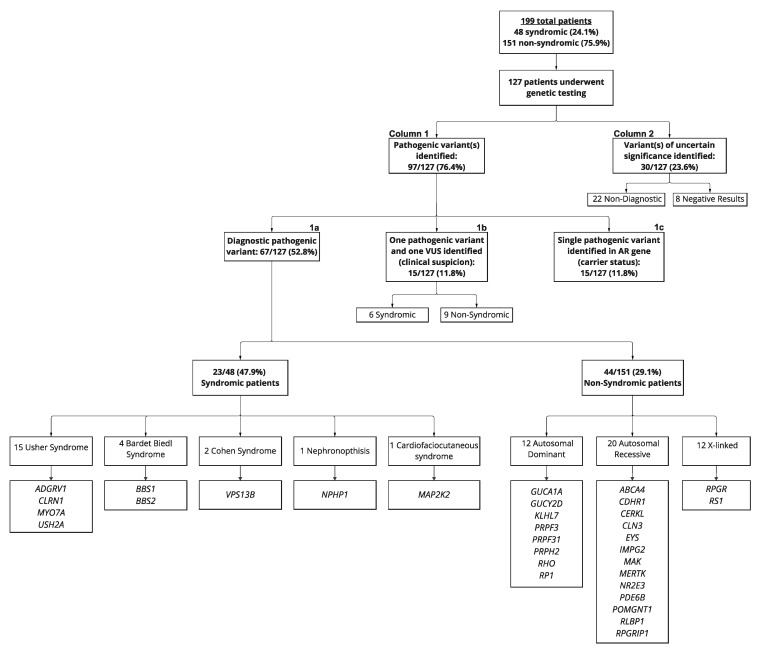
Detailed summary of the patient cohort who underwent genetic testing.

**Figure 3 ijms-24-10895-f003:**
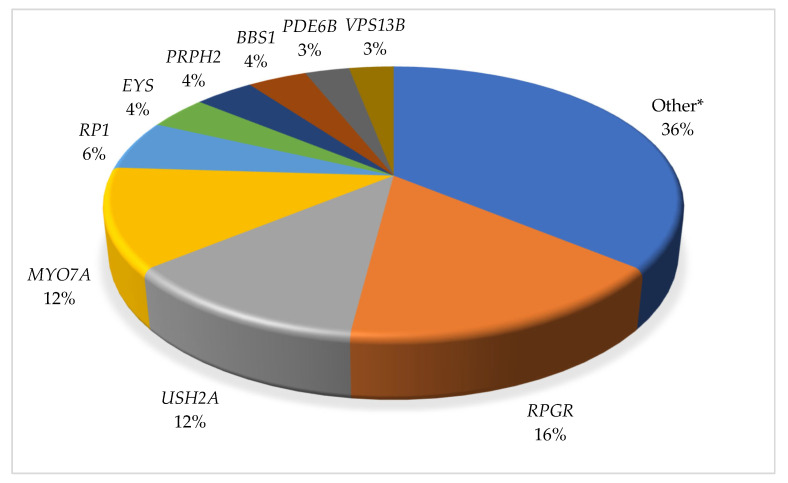
Distribution of causative RP genes identified in the patient cohort. * The ‘Other’ category includes cases in which only a single patient had diagnostic findings in a particular gene. Please refer to Figure 2 for the full list.

**Table 1 ijms-24-10895-t001:** Patient Cohort Demographic Information.

Age at Which the Patient Reported RP Eye Symptoms			Total Cases	
Before the age of 10			61	
Between the ages of 10–19			32	
Between the ages of 20–40			34	
After the age of 40			17	
Unknown			55	
Visual Acuity				
Snellen	≥20/40		<20/40–≥20/80	<20/80
(LogMAR)	(≤0.3)		(>0.3–≤0.6)	(>0.6)
Right eye	93		31	74
Left eye	99		19	80
EZ width	<1500 μm		1500–3500 μm	3501–6000 μm
Right eye	134		9	39
Left eye	134		9	39
FAF Findings	Normal	Macula ring of hypo-AF	Peripheral hypo-AF	Macula ring of hyper-AF
Right eye	4	69	181	91
Left eye	2	68	182	92

**Table 2 ijms-24-10895-t002:** Patient genetic panel utilization and diagnostic yield rate.

Gene Panel *	Nº Patients Utilized	Nº Diagnostic Result	Diagnostic Yield Rate (%)
Invitae Laboratory	71	31	43.7
UMN NGS	22	16	72.7
PreventionGenetics	12	6	50
Blueprint Genetics	9	8	88.9
Other	13	6	46.2

* Each panel may range from 1 to >300 genes tested.

## Data Availability

Not applicable.

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
