# Peer review of "Clinical Characteristics and Genetic Variants of a Large Cohort of Patients with Retinitis Pigmentosa Using Multimodal Imaging and Next Generation Sequencing"

_ijms, 2023, doi:10.3390/ijms241310895_

Round 1

Reviewer 1 Report

The authors describe a sizeable cohort of retinal dystrophy cases and the genetic analysis of a large proportion of this group of patients.

I find it difficult to value the novelty of the ophthalmological data, so I will focus on the genetic data.

The diagnostic ‘solve rate’ is as can be expected for gene-panel or individual gene analysis.

Some elements are missing:

In methods (supplementary data?), details of which genes have been sequenced given the different test labs should be given. As at least one deep-intronic variant was found in USH2A, also list which deep-intronic regions were screened.

The variant data representation should be improved:

1.       Employ HGVS variant nomenclature rules

2.       Splice site variants for which there is no literature data on their effect, the protein notation should be p.(?)

3.       For non-canonical splice site variants, provide SpliceAI or Alamut to provide splice defect predictions

4.       Upload all cases and their variants in the respective LOVD databases. This can be done before the paper is accepted. The data will only become visible in the public domain after acceptance of the paper.

5.       For cases with autosomal recessive variants, provide an identifier number of the proband and the two alleles (cDNA notation and protein notations) next to each other or in consecutive lines. If needed in a supplemental table.

6.       For missense variants, provide in silico predictions on pathogenicity using a few tools (e.g. REVEL, CADD, PhyloP).

7.       For deletions, insert the proper notations. If they are not new, they can be found in the respective LOVDs. For example, for the EYS exon 32-33 deletion, I found this notation in www.lovd.nl/EYS, that considers that the start and end of this deletion are in introns 31 and 33: c.(6424+1_6425-1)_(6725+1_6726-1)del. The predicted protein defect then is: p.(Asp2142Alafs*14)

8.       Provide the ACMG classification for each allele or combination of alleles. In the presence of a trans Pathogenic or Likely Pathogenic allele, a VUS sometimes will change to Likely Pathogenic. Was this considered?

- Indicate which of the variants are novel. This can be looked up in the respective LOVDs and ClinVar

The proband with an ABCA4 variant (p.(Gly1961Glu)) I presume carries this variant in a homozygous manner? If so, it cannot explain an RP phenotype. RP or panretinal CRD can be diagnosed in persons with two severe ABCA4 variants, but p.(Gly1961Glu) has not been observed, to my knowledge, in a homozygous RP case as it is not a fully deleterious allele.

In more detail:

Use * and not X consistently

Use three-letter amino acid code consistently

Use parentheses in protein notations, unless the protein prediction is based on an in vitro or other splice assay. For example, the USH2A variant c.7595-2144A>G results in a pseudoexon insertion and the protein notation then will be: p.Lys2532Thrfs*56

VPS13B: p.P2001LfxX18 cannot be correct

PDE6B: c.1765dpo cannot be correct

Other small issues:

Look critical at the use of capitals. In the Keywords, but also in the text. For example ‘Cardiofaciocutaneous syndrome’ does not need a capital c.

Bardet Biedl syndrome has no capital s

BBS2 case: list the variants identified and refer to the other study (in legend)

Table 1: Why are the first two age at onset groups in bold? Why is there a separator line below these? Similar, line 168: Fundus Autofluorescence should be fundus autofluorescence. Similar in labels at the bottom of Figure 1.

Line 102: delete ‘s’ at the end of ‘panels’

In Figure 2, arrange the genes (if more than one) in alphabetic order

Lines 201-203 contains grammatical errors

English in general is fine

Author Response

Reviewer 1: 

  1. Comment: The authors describe a sizable cohort of retinal dystrophy cases and the genetic analysis of a large proportion of this group of patients. I find it difficult to value the novelty of the ophthalmological data, so I will focus on the genetic data. 

Answer: In this paper, we show that patients with RP are often evaluated in advanced disease independent of the genetic result. This is significant because current and future gene therapy trial and other treatments request for patients at an earlier disease presentation.

Given the global increase in gene therapy and other clinical trials for RP patients and that we currently are the only institution in the Midwest region of the United States to categorize a large cohort of RP patients into a database, we consider our findings presented in our study to appeal to the IJMS.  

In addition, per our reviewer request, we added more genetic data to strengthen our paper.

  1. comment: The diagnostic ‘solve rate’ is as can be expected for gene-panel or individual gene analysis.

Some elements are missing:

In methods (supplementary data?), details of which genes have been sequenced given the different test labs should be given. As at least one deep-intronic variant was found in USH2A, also list which deep-intronic regions were screened.

The variant data representation should be improved:

  1. Employ HGVS variant nomenclature rules
  2. Splice site variants for which there is no literature data on their effect, the protein notation should be p.(?)
  3. For non-canonical splice site variants, provide SpliceAI or Alamut to provide splice defect predictions
  4. Upload all cases and their variants in the respective LOVD databases. This can be done before the paper is accepted. The data will only become visible in the public domain after acceptance of the paper.
  5. For cases with autosomal recessive variants, provide an identifier number of the proband and the two alleles (cDNA notation and protein notations) next to each other or in consecutive lines. If needed in a supplemental table.
  6. For missense variants, provide in silico predictions on pathogenicity using a few tools (e.g. REVEL, CADD, PhyloP). 
  7. For deletions, insert the proper notations. If they are not new, they can be found in the respective LOVDs. For example, for the EYSexon 32-33 deletion, I found this notation in www.lovd.nl/EYS, that considers that the start and end of this deletion are in introns 31 and 33: c.(6424+1_6425-1)_(6725+1_6726-1)del. The predicted protein defect then is: p.(Asp2142Alafs*14)
  8. Provide the ACMG classification for each allele or combination of alleles. In the presence of a trans Pathogenic or Likely Pathogenic allele, a VUS sometimes will change to Likely Pathogenic. Was this considered?

- Indicate which of the variants are novel. This can be looked up in the respective LOVDs and ClinVar

The proband with an ABCA4 variant (p.(Gly1961Glu)) I presume carries this variant in a homozygous manner? If so, it cannot explain an RP phenotype. RP or panretinal CRD can be diagnosed in persons with two severe ABCA4 variants, but p.(Gly1961Glu) has not been observed, to my knowledge, in a homozygous RP case as it is not a fully deleterious allele.

In more detail:

Use * and not X consistently

Use three-letter amino acid code consistently

Use parentheses in protein notations, unless the protein prediction is based on an in vitro or other splice assay. For example, the USH2A variant c.7595-2144A>G results in a pseudoexon insertion and the protein notation then will be: p.Lys2532Thrfs*56

VPS13B: p.P2001LfxX18 cannot be correct

PDE6B: c.1765dpo cannot be correct

Answer: Thank you for your valuable comment. As recommended, we added a supplemental data table that contains details of which variants were identified in each individual patient written in accordance with current HGVS nomenclature, the heterozygosity of variants, and ACMG variant classifications as assigned by the performing genetic laboratory.

With regards to comments 3, 4, and 6, the data in this database was extracted from genetic test reports from various CAP and CLIA certified laboratories. Therefore, we did not perform specific splice predictions and/or in silico predictions. The CAP/CLIA laboratories would submit their variants to the LOVD and/or ClinVar database as is applicable.

  1. Comment:

- Other small issues:

- Look critical at the use of capitals. In the Keywords, but also in the text. For example ‘Cardiofaciocutaneous syndrome’ does not need a capital c.

Bardet Biedl syndrome has no capital s

BBS2 case: list the variants identified and refer to the other study (in legend)

Answer: This is fixed

Table 1: Why are the first two age at onset groups in bold? Why is there a separator line below these? Similar, line 168: Fundus Autofluorescence should be fundus autofluorescence. Similar in labels at the bottom of Figure 1.

Line 102: delete ‘s’ at the end of ‘panels’

In Figure 2, arrange the genes (if more than one) in alphabetic order

Lines 201-203 contains grammatical errors

Answer: This is fixed

Reviewer 2 Report

Sather III et al. realized a very interesting article describing the “Clinical characteristics and genetic variants of a large cohort of patients with retinitis pigmentosa using multimodal imaging and next generation sequencing”. I consider the manuscript very interesting but, at the same time, I suggest several revisions needed to improve the reliability and the completeness of the paper: 

·      The “Introduction” should be more detailed, especially about syndromic forms of RP. E.g., it lacks the Stargardt Syndrome.

·      The “Methods” section should be divided into several sub-chapters.

·      The “Figure 2” is poor in its quality and should be improved.

·      The “Discussion” section should be more updated and improved, especially about the molecular pathways the found mutated genes are involved in. I suggest adding data related to recent bulk transcriptomics studies investigating the vascular alteration and epigenetic impacts on IRDs. The recent PMID: 30523548 and PMID: 36290689 could represent a substrate able to enforce the role of considered cellular mechanisms.

·      Finally, manuscript requires several English revisions and typos correction.

I suggest the authors to revise the whole manuscript for several mistakes and typos.

Author Response

Reviewer 2: 

  1. Comment: Sather III et al. realized a very interesting article describing the “Clinical characteristics and genetic variants of a large cohort of patients with retinitis pigmentosa using multimodal imaging and next generation sequencing”. I consider the manuscript very interesting but, at the same time, I suggest several revisions needed to improve the reliability and the completeness of the paper: 

The “Introduction” should be more detailed, especially about syndromic forms of RP. E.g., it lacks the Stargardt Syndrome.

The “Methods” section should be divided into several sub-chapters.

The “Figure 2” is poor in its quality and should be improved.

The “Discussion” section should be more updated and improved, especially about the molecular pathways the found mutated genes are involved in. I suggest adding data related to recent bulk transcriptomics studies investigating the vascular alteration and epigenetic impacts on IRDs. The recent PMID: 30523548 and PMID: 36290689 could represent a substrate able to enforce the role of considered cellular mechanisms.

Finally, the manuscript requires several English revisions and typos correction.

Answer:

Thank you for the comments. Different sections of the paper in the introduction, methods and discussion have been re-written in response to these suggestions. These changes are noted using tracking changes. Figure 2 is now improved in quality and has been edited accordingly.

We found it useful that the reviewer provided the below references as substrate for the cellular mechanisms of retinitis Pigmentosa. We referenced the paper by Donato et al (PMID 36290689) in our discussion of the limitations of NGS in our cohort shown starting in line 353.

PMID: 36290689: Donato L, Scimone C, Alibrandi S, Scalinci SZ, Rinaldi C, D'Angelo R, Sidoti A. Epitranscriptome Analysis of Oxidative Stressed Retinal Epithelial Cells Depicted a Possible RNA Editing Landscape of Retinal Degeneration. Antioxidants (Basel). 2022 Sep 30;11(10):1967. doi: 10.3390/antiox11101967. PMID: 36290689; PMCID: PMC9598096.

Scimone C, Donato L, Marino S, Alafaci C, D'Angelo R, Sidoti A. Vis-à-vis: a focus on genetic features of cerebral cavernous malformations and brain arteriovenous malformationspathogenesis. Neurol Sci. 2019 Feb;40(2):243-251. doi: 10.1007/s10072-018-3674-x. Epub 2018 Dec 6. PMID: 30523548.

Reviewer 3 Report

Is any information regarding age of the cohort available? And perhaps age of genetic testing or genetic diagnosis?  Curious if perhaps age could be factor for the majority of the cohort having advanced photoreceptor loss?  As it is noted, 'genetic testing at the initial visit' is important as it  'may enable them to identify clinical trials and potential future treatments at the earliest possible disease stage'

Author Response

Reviewer 3: 

  1. Comment: Is any information regarding age of the cohort available? And perhaps age of genetic testing or genetic diagnosis?  Curious if perhaps age could be factor for the majority of the cohort having advanced photoreceptor loss?  As it is noted, 'genetic testing at the initial visit' is important as it  'may enable them to identify clinical trials and potential future treatments at the earliest possible disease stage'

Answer: Thank you for bringing up a good point. Information regarding age of the cohort is available in our table 1 and the supplemental file. Comments were also made in the results section stating at line 203 and in the discussion starting at line 366. The age of the genetic testing and genetic diagnosis has also been added in the supplemental table.  We agreed as the reviewer mentioned that age is an important factor for the majority of the cohort having advanced photoreceptor loss. We speculate that the late presentation to the eye clinic is that patients were not offered treatment options for RP, so patients for many years did not find the need for follow up. Sponsored genetic testing was not available in the past. NGS Genetic panels have been available  for free only in the past 5 years, basically since the approval of Luxterna, the first gene therapy treatment for RPE65 LCA. Because new gene therapy clinical trials are currently undergoing, we have recently advocate genetic testing. This has made possible to start establishing genotype-phenotype correlations and facilitate diagnosis on our patients. All this has been explain now more explicit within the manuscript.